# On the Copenhagen Interpretation of Quantum Measurement

Michael L. Walker 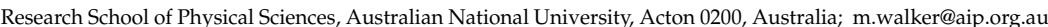

Research School of Physical Sciences, Australian National University, Acton 0200, Australia; m.walker@aip.org.au

**Abstract:** We claim that quantum collapse, as per the Copenhagen interpretation of quantum mechanics, follows naturally from the energetics of measurement. We argue that a realistic device generates an interaction energy that drives a random walk in Hilbert space and generates the probabilistic interpretation of Born.

**Keywords:** quantum measurement; Copenhagen interpretation

## 1. Introduction

### 1.1. What Is the Quantum Measurement Problem?

The discovery of quantum mechanics introduced the perplexing phenomenon of superposition, where an object is found to be in a combination of states, each with a different value, for a given observable. However, measuring a state in superposition does not yield a superposition of values, the measuring device neither displays several values simultaneously nor returns a random sequence of values with repeated measurement. Rather, a measurement returns a single value and subsequent measurements of the same object are consistent with that measurement. Hence the outcomes of future measurements are correlated with those of past measurements, indicating the long-known result [1,2] that any quantum measurement causes a state transition of the measured object. This is commonly described as a "collapse", in which the components not consistent with the measured value are discarded and the object's state is described by a single eigenstate, an eigenstate of the observable's operator. However, to simply discard these other components is a violation of unitarity so a more detailed description of the measurement process is needed.

### 1.2. Models of Quantum Measurement

Quantum mechanics is both universal and linear. The former because no experiment has ever indicated a length or mass scale beyond which quantum mechanics fails [3–6], and the latter because no one has ever found a nonlinear extension of quantum mechanics that agrees with experiment. Furthermore, nonlinear additions to the Schroëdinger equation have long been known to violate relativistic causality [2] in addition to being incompatible with the superposition principle and, therefore, unitarity. However, it is a long-known result of von Neumann [1,2] that it is impossible for quantum mechanics to remain unitary/linear whilst remaining both universal and complete, without invoking either a many-worlds model, such as that proposed by Everett [7], or a classical observables or collapse model.

Classical observables models rely on the somewhat arbitrary division of observables into classical and quantum, where the coupling of the latter to the former via measurement induces a wavefunction collapse in which the other components are discarded. They are typically variations of the Copenhagen interpretation, in which quantum collapse is induced by a classical measuring device, although what constitutes such a device is not well defined.

In objective collapse models, the wavefunctions are constantly collapsing according to a formula contrived to leave too little time to complete a measurement but enough for observed interference and other wave effects. While avoiding the need to define the

measurement, it lacks a mechanism and has been challenged by recent superposition experiments [4].

In the Penrose model of quantum measurement [8], a superposition of positions leads to a superposition of spacetime curvatures, with the resulting energy leading to wavefunction collapse. While this seems difficult to apply to observables in general, our model also uses the energy due to a superposition, although we are more concerned with the energetics of the device than with spacetime.

Indeed, the lack of a convincing collapse mechanism has proven sufficiently vexing for von Neumann to propose, and Wigner to seriously discuss, consciousness as a mechanism (see [9] for a discussion).

Relative state models, also known as Everett's many-worlds interpretation, have the advantage of not needing the introduction of new rules for the collapse, but have nonetheless attracted considerable criticism (for examples see references [10,11]), with particular controversy concerning its reproduction of probabilities (see references [12,13] for examples).

Meanwhile, some authors have abandoned a realistic interpretation of quantum mechanics and considered the issue within either a statistical interpretation [14] or a stochastic model [15].

We propose a mechanism based on conventional physics to explain measurement collapse and the emergence of probabilities in accordance with Born's probabilistic interpretation of the wave function. In Section 2, we discuss the energetics of measurement and demonstrate on these grounds that a classical measuring device cannot typically minimise its energy when measuring a quantum state in superposition. The emergence of Born's probabilistic interpretation from the resulting interaction between the measured state and the measuring device is described in Section 3, while Section 4 summarises our argument and discusses some related issues.

## 2. The Energetics of Measurement

### 2.1. The Energetics of Classical Measurement

As von Neumann noted decades ago [2], a measuring device is a physical system whose ground state depends on a specific property of whatever external object is being measured. (Also see references [16,17] for a discussion of subtle variations.) Since any physical system will seek to minimise its potential energy, the device's minimal energy configuration occurs when its output corresponds to the value being measured, and is in a non-minimal energy state otherwise.

If the device cannot achieve/maintain its minimum energy configuration, one in which its output matches the value(s) found by measurement, then the device-state system remains in an unstable, higher-energy configuration.

### 2.2. The Energetics of Quantum Measurement

Every measuring device is comprised of causally interacting components that propagate information to each other, typically along either atomic bonds or electrically conductive structures. For example, if the measured value is indicated by a needle, the position of its tip, though ultimately determined by the measured quantity, is driven by mechanical forces along the needle at the speed of sound.

Causal interactions are driven by potentials. In the case of a measuring device, the potential is determined by the difference in value between the measured observable and the corresponding output of the device. We henceforth refer to it as the "measurement potential".

Prior to measurement, the device and all its subsystems are in a state independent of the measured observable. Since the measurement potential is generated by the difference in measured values, the device, and its subsystems, evolve towards states corresponding to the components of the measured state in proportion to the magnitude of their coefficients. We might naïvely expect the measuring device to end up in a corresponding superposition but we shall demonstrate that a single-component state has lower energy.

### 2.3. The Measurement Interaction Energy

To study the effect of measurement potential, we give the quantum state being measured by $|\psi(t)\rangle$, where $t$ is the length of time from when measurement commences. It decomposes into the eigenstates $|\psi_i\rangle$ of the operator whose observable is being measured as

$$|\psi(t)\rangle = \sum_i \lambda_i(t) |\psi_i\rangle, \tag{1}$$

where at any given $t$

$$\sum_i \|\lambda_i\|^2(t) = 1. \tag{2}$$

The variation of $\lambda_i(t)$ depends on the specific details of the measuring device and the initial quantum state and is typically unavailable.

Interaction between the device and the measured object cannot be longer than the timescale of the quantum fluctuations or the device will measure a mean value; so, we assume that fluctuations in the $\lambda_i$ are on a timescale much shorter than the relaxation state of the device. We shall simplify the notation henceforth by not indicating time dependence explicitly but instead trusting that it has been sufficiently indicated already.

Remembering from the previous section that the measurement potential is generated by a mismatch between the observables and their measured values, we expect each component $|\psi_i\rangle$ of the wave function to energetically prefer an output of $A_i$ corresponding to its observable value $a_i$, so the measurement energy of each component $|\psi_i\rangle$ receives a contribution from all the other wave function components and is given by

$$M \sum_{j;j\neq i} |\lambda_j|^2 = M \sum_{j;j\neq i} \langle\psi| \lambda_j |\psi_j\rangle, \tag{3}$$

for some large positive real $M$, which is independent of the index $i$. Summing this over all $|\psi_i\rangle$, weighted by their probability amplitudes $\lambda_i$ then yields

$$M \sum_{i,j;j\neq i} \langle\psi| \lambda_j |\psi_j\rangle \langle\psi| \lambda_i |\psi_i\rangle \tag{4}$$

$$= \frac{M}{2} \sum_{i,j;j\neq i} \left( \langle\psi| \lambda_i |\psi_i\rangle \langle\psi_j| \lambda_j^* |\psi\rangle + \langle\psi| \lambda_j |\psi_j\rangle \langle\psi_i| \lambda_i^* |\psi\rangle \right)$$

$$= \langle\psi| \frac{M}{2} \sum_{i,j;j\neq i} \left( \lambda_i \lambda_j^* |\psi_i\rangle \langle\psi_j| + \lambda_j \lambda_i^* |\psi_j\rangle \langle\psi_i| \right) |\psi\rangle$$

$$= \langle\psi| \mathcal{M} |\psi\rangle, \tag{5}$$

where the last line implicitly defines the measurement energy operator $\mathcal{M}$ to be

$$\mathcal{M} \equiv \frac{M}{2} \sum_{i,j;j\neq i} \left( \lambda_i \lambda_j^* |\psi_i\rangle \langle\psi_j| + \lambda_j \lambda_i^* |\psi_j\rangle \langle\psi_i| \right). \tag{6}$$

## 3. The Resultant Random Walk Leads to the Emergence of Born's Probabilistic Interpretation

From the form of the measurement potential operator $\mathcal{M}$, given in Equation (6), we see that it couples between orthogonal components symmetrically, up to a phase, thus leading to transitions of equal likelihood each way. Acting on the wave function over time, this symmetric coupling of the operator $\mathcal{M}$ between all pairs of components drive the wave function to constantly fluctuate randomly towards some components at the expense of others. These fluctuations describe a random walk in the subspace of Hilbert space generated by the superposition's components.

The random walk is restricted by unitarity to unit distance from the origin in Hilbert space so that the sum of the squares of the coefficients is fixed. When it reaches a direc-

tion orthogonal to one of its components that component vanishes and can no longer be transitioned to.

Consider, for example, a superposition of two eigenstates $|a\rangle$, $|b\rangle$ with complex coefficients $A$, $B$, respectively, written

$$A |a\rangle + B |b\rangle, \qquad \|A\|^2 + \|B\|^2 = 1, \tag{7}$$

any change in $\|A\|^2$ is exactly offset by a corresponding but opposite change in $\|B\|^2$, so that $|a\rangle$, $|b\rangle$ may be said to be randomly exchanging $(\delta\|A\|)^2$ for $(\delta\|B\|)^2$ in parallel to Huygen's statement of the gambler's ruin problem [18]. This problem considers two gamblers repeatedly wagering equal amounts in a fair game until one of them goes bankrupt, and asks each one's chances of winning overall. The long-known solution is that the chances of each are in proportion to the amount of money they started with.

Translating back to quantum measurement, it follows that any component's probability of being chosen is the square of its own coefficient, in agreement with Born's probabilistic interpretation of the wavefunction. To generalise this argument from two to arbitrary numbers of components consider each component in turn competing with the combination of all the others. Hence, each component is found with a likelihood given by the square of its coefficient.

## 4. Discussion

### 4.1. A Natural Definition of a Measuring Device/System

It would seem highly contrived indeed for the apparent collapse of a quantum superposition into a single eigenstate to occur only in modern physics laboratories. A satisfying model must surely accommodate and expect such events under natural conditions prior to modern physics or even the evolution of conscious beings. A true understanding of quantum measurement clearly requires that we can characterise the physical systems that induce measurement collapse. In this paper, we have argued that sufficient conditions are:

1. That the lowest energy state be restricted by the value of the measured observable;
2. That the interaction between the measured state and the measuring system be shorter than the timescale of fluctuations of the superposition.

Any device capable of measuring a superposed observable must fulfil the first two of these criteria. The first condition is true of any measuring device by definition, and if the second condition does not hold then the device will only return a mean value of the superposition.

### 4.2. Natural Emergence of Quantum Collapse and Born's Interpretation

Other collapse models require the introduction of new rules to govern quantum collapse. The original Copenhagen interpretation has the classical measurement with a macroscopic device as a trigger, without clearly defining what it means for a device to be macroscopic. Intuitively, it is taken to mean something on a similar length or mass scale to our everyday experience but no length or mass scale has been identified at which physical systems seem to transition to being macroscopic. Indeed, there is mounting evidence that there are no such scales with interference patterns observed with molecules tens of thousands of times more massive than the hydrogen atom [4,5] and EPR phenomenology demonstrated via satellite [3]. Spontaneous collapse models, on the other hand, simply assert that collapses are constantly occurring according to empirically determined rules.

Our claim is that collapse under conditions universal, but not exclusive, to measurement in the sense of von Neumann [2] follows from simple energetics. The Copenhagen interpretation then emerges as an effective model, and we have plausibly argued that Born's probabilistic interpretation then follows from a random walk and the gambler's ruin problem. At the same time, the natural occurrence of measurement-induced collapse through simple conventional physics leaves no incentive for additional edgy phenomena, like consciousness or the many worlds of Everett's model.

*4.3. Other Considerations*

4.3.1. Quantum Non-Locality

How do the apparently local laws of classical physics emerge from non-local quantum mechanics? EPR phenomenology has been demonstrated over distances as long as 1200 km via satellite [3]. We have made no attempt to address the issue in this paper, but the question arises as to whether our energy-driven mechanism of quantum collapse is reconcilable with quantum non-locality.

One might object that the collapse is triggered through interaction with a classical, localised device. However, while this is true, the final state and measured value arise from the measured state's random walk, which is driven by its self-interaction. The measured state is already non-local in an EPR-relevant situation so, without pretence of rigour, it should not be surprising if its self-interaction were also. This could be argued to support the claim by Aharanov et al. [16,17] that the wave function is a real entity. The notion is contentious, with other authors arguing to the contrary [19].

Alternatively, the random directions taken in the state's random walk might also be driven by non-local vacuum fluctuations.

4.3.2. Is the Walk Random?

Our model predicts Born's statistical interpretation of the wave function through the mechanism of a random walk generated by self-interaction. However, the sequence of transitions still appears to be random, and is certainly not predicted by this work. Whether it is truly random as conventionally believed or driven by unknown or inherently unknowable effects remains an open question.

**5. Conclusions**

We have argued that the wave function collapse of the Copenhagen interpretation occurs naturally in quantum measurement due to conventional energetics. The resulting interaction energy leads to the quantum state performing a random walk in Hilbert space in which one of the components is selected. The fact that each component's probability of being selected is given by Born's interpretation follows naturally from the gambler's ruin problem and this is important support for our model. We therefore find that measurement collapse occurs both naturally and inevitably from the Schroëdinger equation without additional rules or phenomena.

**Funding:** This research received no external funding.

**Data Availability Statement:** New data were created for this article and this article contains no data.

**Acknowledgments:** The author thanks S.O. Bilson-Thompson and F. Perrine-Walker for their constructive criticisms of the manuscript.

**Conflicts of Interest:** The author declares no conflicts of interest.

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
