# Peer review of "On the Copenhagen Interpretation of Quantum Measurement"

_universe, doi:10.3390/universe10030113_

Round 1

Reviewer 1 Report

Comments and Suggestions for Authors

Dear Author

you wrote: "Interaction between the device and the measured object cannot be longer than the timescale of the quantum fluctuations or the device will measure a mean value."

I do not understand how a single quantum measurement is able to give me a mean value.

You wrote: "We therefore assume henceforth that fluctuations in the λi are on a timescale much shorter than the relaxation state of the device, so they may be taken as having an effectively constant amplitude on this timescale."

So, you exclude time from your model completely, is'n it?

But you declared:" We might therefore expect the measuring device to end up in a corresponding superposition but we shall demonstrate that a single-component state has lower energy." It means that you should find time evolution of an operator which describe the measuring device (or measument procedure).

As for equation (3), I do not see what namely parameter labeled by index "i"  in both side of the equation. Hence, I do not understand futher procedure of getting operator M.

Any way, you called the operator M as "measurement energy operator". Even if you get an operator M in any shape, I do not see an equation which describe an evolution in time the operator or any idea about what should be done to get result of the measurement using the operator.

You mentioned that the operator "leading to transitions" (lines 120-122),   but for me it is not enough to see what is the result of the measurement.

Common comment:

When you introduce new approach in physics, describe the  approach as detailed, as possible.

Reviewer 2 Report

Comments and Suggestions for Authors

Personally I like the suggested connection with the gambler's ruin problem. Although the problem of measurement has been considered many times before, this exact explanation seems a little new to me, and thus I recommend this paper for immediate publication in MDPI.

Author Response

No responses were required for this reviewer.

Reviewer 3 Report

Comments and Suggestions for Authors

This paper tries to explain how the collapse of the wavefunction occurs in the Copenhagen interpretation of quantum mechanics. The author proposes a dynamical scheme that involves the self-interaction of the system during the measurement, the energetics of which lead it to perform a random walk. The author thereby argues that the outcome can be viewed as that of a classical random walk generated by the energetics of the quantum processes during the act of measurement.

The problem addressed is a truly difficult one that has been addressed by many authors before and will doubtless be addressed by many in the future. Although some of the assumptions made by the author can be questioned and debated, the paper is not obviously wrong. Thus I would recommend publication, for the discussion it might generate.

Author Response

No responses were required of this reviewer.

Round 2

Reviewer 1 Report

Comments and Suggestions for Authors

Dear Author

I do not see clear description of operator M. The operator exists in both side of formula (3), (4) and (5), and it is strange. I would like to see formula with operator M in left side and the result of the operator action in right side of the formula.

Author Response

The 'M' in equations (3,4) is in a different font to that in equation (5), which is calligraphic. As indicated in the text the former represents a large real constant while the latter represents an operator. We see now that the differences may not be obvious in the text so we have now added equation (6) to define the operator calligraphic 'M' explicitly.

Round 3

Reviewer 1 Report

Comments and Suggestions for Authors

Dear Author

I did not find clear description of operator M.

Author Response

Dear reviewer,

I consider subsection 2.3 to be a step-by-step construction of operator M (not to be confused with constant M which us in a different script). I have slightly reworded the paragraph leading in to equation (3) which may help. If the reviewer would kindly indicate which step or steps are difficult to follow I shall endeavour to clarify them further.

sincerely

Round 4

Reviewer 1 Report

Comments and Suggestions for Authors

Dear Author,

I can not find clear description of physical parameter M in formula (3). You mentioned that it is "large positive real M" only. But it is not enough to introduce the parameter into your model.